

# The role and underlying mechanisms of irisin in exercise-mediated cardiovascular protection

Wenhuang Guo[1,*], Jianwei Peng[1,*], Jiarui Su[1], Jingbo Xia[1],
Weiji Deng[1], Peilun Li[1], Yilin Chen[1], Guoqing Liu[1], Shen Wang[1] and
Junhao Huang[1,2]

[1] Guangdong Provincial Key Laboratory of Physical Activity and Health Promotion, Scientific
Research Center, Guangzhou Sport University, Guangzhou, China
[2] Dr. Neher's Biophysics Laboratory for Innovative Drug Discovery, Macau University of Science
and Technology, Macau, China
* These authors contributed equally to this work.

## ABSTRACT

Irisin, a product of the post-translational processing of fibronectin type III
domain-containing protein 5 (FNDC5), is a novel myokine which is upregulated
during exercise. This hormone not only promotes the transformation of white
adipose tissue into a brown-fat-like phenotype but also enhances energy expenditure
and mitigates fat accumulation. Its role is crucial in the management of certain
metabolic disorders such as diabetes and heart disease. Of note, the type of exercise
performed significantly affects blood irisin levels, indicating the critical role of
physical activity in regulating this hormone. This article aims to summarize the
current scientific understanding of the role of irisin and the mechanisms through
which it mediates cardiovascular protection through exercise. Moreover, this article
aims to establish irisin as a potential target for preventing and treating cardiovascular
diseases.

## INTRODUCTION

Cardiovascular diseases (CVDs) are a clinical burden globally and statistics from a
worldwide cohort indicate that the incidence of CVD is 57.2% in women and 52.6% in men
with a median age of 54.4 years for both groups, respectively (*Magnussen et al., 2023*). In
China, CVDs have been intricately influenced by a confluence of demographic changes,
environmental influences, lifestyle choices, and accessibility of healthcare services. The
quest to mitigate significant personal suffering as well as societal and familial burdens
associated with CVDs has become an important objective in contemporary medical
research (*Zhao et al., 2019*). Among the various strategies being explored, exercise training
has emerged as a major preventive and therapeutic intervention for CVDs. Indeed, it has
been widely acknowledged for its noninvasive nature and profound benefits in CVD
management (*Chen et al., 2022a*). Studies have revealed that exercise-induced myokine
irisin has an efficient therapeutic effect on several metabolic diseases such as type two

Corresponding authors
Shen Wang, wangs@gzsport.edu.cn
Junhao Huang,
junhaohuang2006@hotmail.com

diabetes (*Lin et al., 2021*), insulin resistance (*Balakrishnan & Thurmond, 2022*), non-alcoholic fatty liver disease (*Zhang et al., 2013*), and CVDs (*Fu et al., 2021*). The expression of irisin is intricately linked to the activation of peroxisome proliferator-activated receptor γ coactivator 1α (PGC1α) (*Bostrom et al., 2012*; *Kelly, 2012*), and exercise has been shown to stimulate irisin secretion, thereby promoting cardiovascular health (*Liu, Wei & Wang, 2022*). The present review delves into the multifaceted role of irisin, a myokine triggered by physical activity, in the realm of cardiovascular protection. This article discusses the common pathways that connect exercise-induced irisin production with CVD mitigation, thereby offering insights into the potential mechanisms and interconnections that underlie this phenomenon.

## SURVEY METHODOLOGY

Original data and information for this review was retrieved from journal articles in PubMed, Google Scholar, and Elsevier databases using the keywords "irisin and exercise" or "irisin and cardiovascular diseases".

### Discovery and properties of irisin

In 2012, *Bostrom et al. (2012)* revealed that skeletal muscle can release PGC1α after exercise. In addition, PGC1α was shown to regulate energy metabolism and promote several processes, such as mitochondrial biogenesis, skeletal muscle fiber type switching, anti-oxidation, angiogenesis, and others (*Bennett, Latorre-Muro & Puigserver, 2022*; *Fujiwara et al., 2023*; *Lira et al., 2010*). PGC1α in muscle tissue was demonstrated to increase the expression of the membrane protein fibronectin type III domain-containing protein 5 (FNDC5), which upon cleavage and release, generates a novel hormone termed irisin (*Norheim et al., 2014*). In the skeletal muscle, the hormone irisin can promote myogenesis and inhibit muscle atrophy *via* autocrine and/or paracrine mechanisms (*Reza et al., 2017*; *Rodríguez et al., 2015*). Of note, irisin drives subcutaneous white adipose tissue browning and body thermogenesis (*Bostrom et al., 2012*). It mediates its effects on adipose tissue *via* αV integrin receptors (*Kim et al., 2018*). In the adipose tissue of patients with obesity, the expression of the gene encoding irisin precursor (FNDC5) is decreased and that of integrin αV integrin receptor is increased, suggesting an attempt to overcome irisin deficiency (*Frühbeck et al., 2020*; *Moreno-Navarrete et al., 2013*).

A study showed that circulating irisin levels were positively correlated with the body mass index (*Huh et al., 2015*) and serum glucose level (*Xiong et al., 2015*). By contrast, irisin levels were negatively correlated with age, insulin, cholesterol, obesity, and adiponectin level (*Huh et al., 2012*; *Moreno-Navarrete et al., 2013*). Additional experiments confirmed the potential mechanisms underlying irisin-mediated effects on body thermogenesis, adipose tissue remodeling, and obesity progression. Irisin stimulates the p38 MAPK and ERK signaling pathways, and initiates the browning process of the white adipose tissue. This transformation boosts energy expenditure, enhances glucose tolerance, and ameliorates insulin resistance (Fig. 1) (*Waseem et al., 2022*; *Zhang et al., 2014*). In addition, brown adipose tissues dissipate energy produced by the oxidation of body

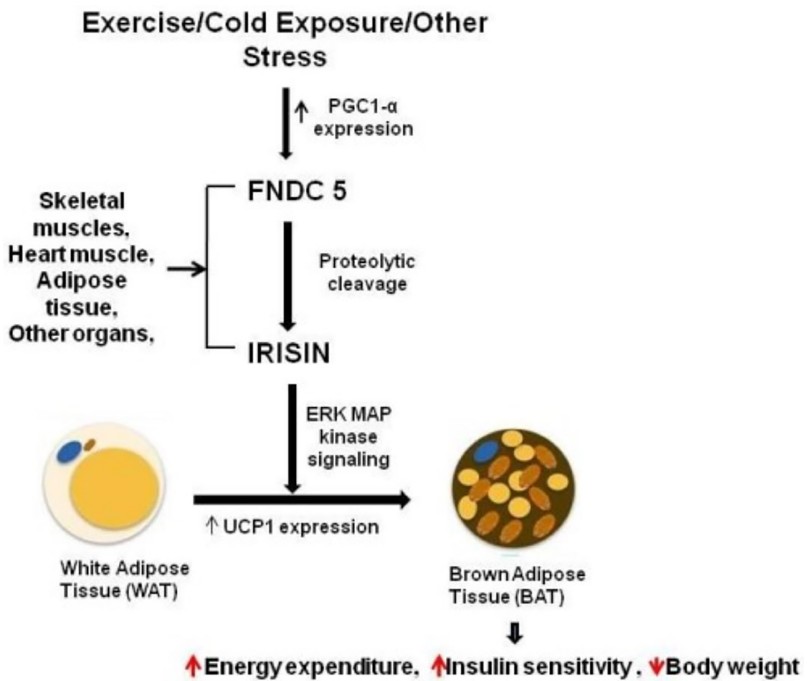

**Figure 1 Irisin secretion and its role in fat browning.** Reference from *Waseem et al. (2022)*.

thermogenesis *via* uncoupling protein 1 (*Erden et al., 2016*; *Grzeszczuk, Dzięgiel & Nowińska, 2024*). The PGC1α-FNDC5-irisin axis was established as the theoretical basis for the energy metabolism mechanism (*Kelly, 2012*).

## Irisin protein structure and expression

The precursor of irisin is FNDC5, a vital muscle protein with a signal peptide, two fibronectin type III domains, a transmembrane segment, and a cytoplasmic tail. Irisin, a PGC1α-regulated myokine, is proteolytically cleaved from FNDC5 and secreted to modulate metabolism (*Rabiee et al., 2020*). This structural organization is important for the function of FNDC5, particularly in metabolism regulation and exercise physiology (*Waseem et al., 2022*). The N-terminal fragment of FNDC5 is located in the cytoplasm, and its extracellular portion is cleaved *via* protein hydrolysis to produce irisin (*Bostrom et al., 2012*; *Korta, Pocheć & Mazur-Biały, 2019*; *Rabiee et al., 2020*). Both FNDC5 and irisin were first discovered in the skeletal muscle and serum of humans, rabbits, and mice (*Hofmann, Elbelt & Stengel, 2014*). In mammals, the amino acid sequence of irisin is highly conserved (*Ning, Wang & Zhang, 2022*), with nearly 100% homology between humans and mice, notably higher than the 90% for glucagon, 85% for insulin, and 83% for leptin (*Bostrom et al., 2012*). Irisin has also been detected in the brain and skin of rats, with residual levels observed in their liver, pancreas, spleen, stomach, and testis (*Aydin et al., 2014b*). While FNDC5 mRNA is abundant in the pericardium of humans, low levels have been detected in the kidney, liver, lung, neuron, and adipose tissue (*Flori, Testai & Calderone, 2021*; *Kim et al., 2017*; *Zhang et al., 2022*). In addition, irisin can be detected in human cerebrospinal
fluid, saliva, and breast milk (*Aydin et al., 2013*; *Piya et al., 2014*; *Pomar, Sánchez & Palou, 2020*). As previously mentioned, in humans, circulating irisin levels were positively correlated with insulin resistance, fasting blood glucose, body mass index, muscle mass, and fat-free mass but negatively correlated with age, insulin, cholesterol, adiponectin, and triglycerides (*Aydin et al., 2014a*; *Huh et al., 2012*; *Xiong et al., 2015*). Meanwhile, exercise has been shown to induce the secretion of irisin and promote the expression of its precursor FNDC5 (*Bostrom et al., 2012*; *Rabiee et al., 2020*). Other physicochemical factors, such as starvation, frigidity, high temperature, metformin, and follistatin, can also promote irisin expression (*Aydin et al., 2013*; *Lee et al., 2014*; *Lin et al., 2021*; *Liu, Wei & Wang, 2022*; *Luo et al., 2023*; *Roca-Rivada et al., 2013*).

## Protective role of irisin in the cardiovascular system

Since its discovery, the role of irisin in obesity, type two diabetes, metabolic diseases, nephropathy, and CVDs has been a focal point of research. In particular, the association between irisin and CVDs has received widespread attention. Studies have shown that exercise can stimulate the secretion of irisin in various parts of the body *via* multiple mechanisms and this secretion is crucial for protecting cardiovascular function and preventing CVDs (Fig. 2) (*Qin et al., 2022*). At present, the function of irisin in the body, particularly how it is induced by exercise, has become a hot topic of research. For example, acute exercise significantly increases irisin levels in the blood, whereas long-term exercise helps improve its metabolic dynamics (*Ma et al., 2021*). Irisin is not only strongly correlated with CVDs but also has the potential to serve as a biomarker for CVD diagnosis. Circulating irisin levels are negatively correlated with several risk factors for cardiovascular health, such as hyperglycemia, triglycerides, visceral adiposity, and extramyocellular lipid deposition (*Kurdiova et al., 2014*). By activating the AMPK-eNOS signaling pathway, irisin improves vascular endothelial dysfunction and lowers inflammatory factor levels in the blood, thereby protecting the vascular endothelium (*Fu et al., 2021*). Additionally, irisin promotes vascular endothelial cell proliferation and inhibits oxidative stress and inflammatory responses, thereby improving the vascular endothelial function in diabetic mice (*Han et al., 2015*). In terms of therapeutic applications, irisin has been demonstrated to alleviate cardiac dysfunction and ventricular dilation, reduce the infarct area of myocardial infarction (MI), and decrease MI-induced fibrosis. The molecular mechanisms behind these therapeutic effects involve angiogenesis and ERK signaling pathway activation in endothelial cells (*Liao et al., 2019*). Thus, irisin plays multiple roles in cardiovascular health, and its study may offer new perspectives and potential targets for the prevention and treatment of CVDs.

## Role of irisin in myocardial protection

The cardiovascular system has significant energy demands. A remarkable feature of CVDs is myocardial metabolism disorders under pathological circumstances. The pathological remolding of the heart correlates with glucose and aliphatic acid metabolism. In this context, high irisin expression in human and rat hearts and its role in improving glucose tolerance and insulin sensitivity suggest that it can ameliorate myocardial metabolic

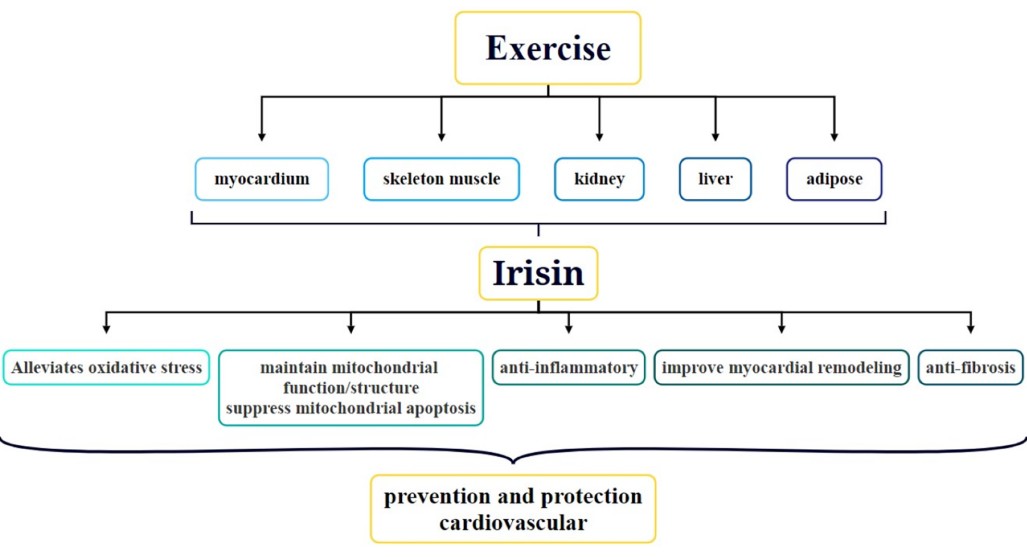

**Figure 2 Role and underlying mechanisms of irisin in exercise-mediated cardiovascular protection.**

disorders (*Flori, Testai & Calderone, 2021*). Studies have revealed low circulating irisin levels in patients with diabetic cardiomyopathy (*Lin et al., 2021*) and heart failure (HF) with reduced aerobic exercise performance (*Lecker et al., 2012*). In addition, *Kuloglu et al. (2014)* demonstrated decreased circulating irisin levels in adrenaline-induced myocardial infarction in rats, indicating that reduced irisin levels correlate with an activated sympathetic nervous system and gradually lower serum irisin levels have a diagnostic significance in MI. Another study that measured irisin levels in mice with cardiac hypertrophy suggested irisin as a diagnostic marker and modulator of cardiac hypertrophy (*Yu et al., 2019*). In patients with acute HF higher serum irisin levels were associated with increased mortality, indicating that serum irisin is a predictive biomarker of 1-year all-cause mortality in acute HF patients. This finding was attributed to the increased circulating irisin levels in patients at risk of CVDs or major adverse cardiovascular events being a manifestation of irisin resistance (*Shen et al., 2017*). Meanwhile, the decoupling effects of irisin can lead to ATP loss, resulting in a poor cardiovascular prognosis (*Aydin et al., 2014a*). *Sobieszek et al. (2020)* suggested that combined analysis of several non-invasive markers, such as irisin, albumin, and inflammatory markers, could offer novel opportunities for improving clinical outcomes in the management of cardiac cachexia in patients with chronic heart failure. Although the effects of irisin on myocardial function have been identified, the underlying molecular mechanisms are unknown. Therefore, additional studies are warranted to provide further evidence.

## The function of irisin in vascular protection

CVDs are highly correlated with endothelial dysfunction (*Chen et al., 2022b*; *England et al., 2018*), whose main feature is the abnormal regulation of vascular tone and abnormal expression of adhesion molecules. Certain pathological factors, such as diabetes mellitus, hyperlipidemia, and smoking can induce vascular endothelial dysfunction

(*Md Salleh et al., 2021*; *Yang et al., 2016*; *Zhu et al., 2015*). *In vitro* experiments have shown that irisin suppressed oxidative/nitrative stress by inhibiting the activation of protein kinase C-β/nicotinamide adenine dinucleotide phosphate oxidase and nuclear factor kappa-B/inducible nitric oxide synthase pathways in human umbilical vein endothelial cells (*Zhu et al., 2015*). Regarding *in vivo* studies, *Chen et al. (2022b)* used irisin to disrupt atherosclerosis in apolipoprotein E knock-out (ApoE$^{-/-}$) mice induced with nicotine for 8 weeks and showed that irisin exerted a reversal effect on intimal thickening caused due to smoking-induced atherosclerosis in these mice through the integrin αVβ5 receptor. Another study demonstrated that irisin could potentially have a significant impact on protection against endothelial damage and reducing atherosclerosis alleviation in ApoE$^{-/-}$ diabetic mice (*Lu et al., 2015*). In addition, the serum irisin level of patients with type 2 diabetes positively correlated with endothelium-dependent vasodilation and low irisin level was the individual pathogenic factor of vascular endothelial disorders (*Chi et al., 2022*; *Hou, Han & Sun, 2015*). Meanwhile, *Lu et al. (2015)* demonstrated *via in vivo*, *ex vivo*, and *in vitro* experiments that the protective effect of irisin on the endothelium was achieved by activating the AMPK-PI3K-Akt-eNOS signaling pathway, which affected the functionality and quantity of endothelial progenitor cells. Results from a study conducted by the present group revealed irisin-induced endothelium-dependent vasodilation through the activating of the transient receptor potential vanilloid subtype 4 pathway and promotion of Ca$^{2+}$ influx in rat mesenteric artery endothelial cells (*Ye et al., 2018*). Together, these results demonstrated that irisin levels are closely correlated with vascular endothelial function. Irisin can affect the function of vascular endothelial cells by regulating the inflammatory response of vascular endothelium, nitric oxide production, and the quantity and activity of endothelial progenitor cells. However, the underlying regulatory mechanisms of these processes need to be further investigation.

CVD, including coronary heart disease, hypertension, HF, and stroke, are the leading causes of morbidity and mortality worldwide, accounting for nearly 30% of the total deaths worldwide (*Fu et al., 2021*). Atherosclerosis, a chronic progressive vasculitis disease with primary clinical manifestations of ischemic heart disease, ischemic stroke, and peripheral arterial disease, is a highly complex multifactorial disease whose pathophysiologic events encompass thrombus formation, endothelial dysfunction, lipid infiltration, oxidative stress, and vascular inflammation (*Cheng et al., 2021*; *Herrington et al., 2016*). Severe coronary arteritis may lead to coronary artery disease (CAD). Many studies have focused on how to treat and prevent coronary atherosclerosis. In this regard, irisin has been shown to be an independent predictive indicator of CAD severity in patients with stable CAD (*Efe et al., 2017*). *Anastasilakis et al. (2017)* measured irisin levels in patients with CAD and MI and found that circulating irisin levels were lower in these patients than in controls and were associated with the degree of coronary stenosis, suggesting that its secretion is regulated per the sufficiency of blood supply to the heart muscle. Irisin was also found to enhance cell viability, migration, and tube formation *in vitro*; its proangiogenic effect on endothelial cells treated with oxidized low-density lipoprotein is mediated through the activation of the AKT/mTOR/S6K1/Nrf2 pathway (*Zhang, Xu & Jiang, 2019*). These studies indicate that serum irisin can improve atherosclerotic diseases, and that

abnormal serum irisin levels can potentially be used as a biomarker to predict the occurrence of coronary atherosclerosis.

## Protective role of exercise-induced irisin in the cardiovascular system

Although irisin can be used as a biomarker of CVDs and is induced by exercise, the effect of exercise-induced irisin secretion on CVDs is unclear. Skeletal muscle, the largest organ in the human body, accounts for approximately 40% of body weight and plays a key role in determining the basal metabolic rate. Beyond its role in muscle contraction, it can synthesize and secrete several myokines (*Febbraio & Pedersen, 2005*; *Gheit et al., 2022*; *Lee et al., 2015*), including irisin, IL-6, IL-8, IL-15, BDNF, CNTF, VEGF, and FGF21. These myokines are hormones produced by skeletal muscle tissue that may act as the molecular mediators of the systemic effects of exercise, thereby influencing other organs (*Schnyder & Handschin, 2015*). FNDC5/irisin-regulated signaling pathways have obvious exercise inductivity (*Liu et al., 2022*). Irisin has been shown to accelerate adipose consumption and protect the cardiovascular system. Thus, inducing irisin secretion could be a potential strategy for preventing and treating CVDs. Previous studies have shown that different exercise forms can affect serum irisin levels (*D'Amuri et al., 2022*; *Kim & Kim, 2018*). In the following sections, the effects of acute and long-term exercise on irisin secretion will be addressed. Serum irisin levels are considered as the potential biomarkers or predictors of CVDs because they are negatively correlated with the risk factors of cardiovascular health and may exert protective effects on the cardiovascular system through various biological pathways (*Fu et al., 2021*). Thus, the impact of exercise-induced irisin secretion on CVD prevention and treatment and the possible mechanisms of the protective role of exercise-regulating irisin on patients with CVD will also be summarized in this section.

## Effect of acute exercise on irisin secretion

Many acute exercise regimens, such as swimming, whole-body vibration, and resistance exercise (RE), have been reported to elicit an increase in irisin levels immediately after exercise (*Huh & Mantzoros, 2015*). A meta-analysis, which confirmed this association, demonstrated a significant rise in irisin levels in adults following acute exercise (*Fox et al., 2018*). When young subjects engaged in moderate-intensity continuous training (MICT) and high-intensity intermittent training (HIIT), the latter was found to induce a higher peak in irisin levels (*Colpitts et al., 2022*). This increase was particularly pronounced in healthy young individuals, suggesting a potential association between exercise intensity and irisin response. In another study involving a 50-min exhaustive exercise at approximately 80% of maximal oxygen uptake ($VO_{2max}$), serum irisin levels were significantly increased from baseline measures (*Qiu et al., 2018*). Likewise, in a group of older adults categorized by physical fitness levels, those with higher fitness had greater baseline irisin levels, although the levels after exercise did not differ significantly (*Bizjak et al., 2021*). However, not all studies reported an increase in irisin levels after exercise. A study involving 1 h of low-intensity training at 50% $VO_{2max}$ observed no change in serum irisin levels (*Pekkala et al., 2013*). In addition, an *in vivo* study on mice found no alteration in circulating irisin levels after an acute swimming session, highlighting inconsistencies in

**Table 1 Effects of exercise on irisin secretion.**

| Model | Age | Exercise intervention style | Exercise frequency | Irisin detection methods and changes | Author (year) |
|---|---|---|---|---|---|
| Human | 17.14 ± 1.66 and 16.27 ± 2.05 years | MCI[1] | 1 time | Blood; ↑; Compared with baseline, $p = 0.049$ | Colpitts et al. (2022) |
| | | HIIT[2] | 1 time | | |
| Human | 27.4 ± 3.8 and 24.7 ± 2.5 years | Acute exercise 80% peak $VO_2$ | 1 time | Blood; ↑; Compared with baseline, $p < 0.05$ | Qiu et al. (2018) |
| Human | 31.2 ± 5. 3 and 29.9 ± 6.4 years | Aerobic exercise | 1 time | Blood; –; Compared with post exercise, no significance | Lagzdina et al. (2020) |
| Human | 74.4 ± 5.7 and 76.1 ± 5.2 years | High physical fitness (HPF) | 16 weeks | Blood; ↑; HPF compared with LPF in basal value of irisin, $p = 0.0195$ | Bizjak et al. (2021) |
| | | Low physical fitness (LPF) | | | |
| Human | 37.2 ± 9.1 and 40.1 ± 7.0 years | MICT | 3 times/week | Blood; ↑; Compared with baseline, $p < 0.05$ | D'Amuri et al. (2022) |
| | | HIIT | 12 weeks | | |
| Human | 62.3 ± 3.5 years | Resistance exercise | 2 times/week | Blood; ↑; Compared with pre-training, $p < 0.01$ | Zhao et al. (2017) |
| | | | 12 weeks | | |
| Human | 46–60 years | Resistance exercise | 3 times/week | Blood; ↑; Compared with control, $p < 0.001$ | Amanat et al. (2020) |
| | | Aerobic exercise | 12 weeks | | |
| | | Combined exercise | | | |
| Human | 68.0 ± 6.2 and 66.5 ± 5.0 years | Resistance exercise | 2 times/week | Blood; ↑; Compared with control, no significance | Tibana et al. (2017) |
| | | | 16 weeks | | |
| Human | 67.7 ± 5.8 years | FTBFR | 3 times/week | Blood; ↑; Compared with control, no significance | Pazokian, Amani-Shalamzari & Rajabi (2022) |
| | | FT | 6 weeks | | |
| Mouse | 16 weeks | Swimming | 1 time | BAT; ↑; Compared with control, $p = 0.0583$ | Cho et al. (2021) |
| Mouse | 14 months | Aerobic exercise | 5 days/week | Blood; ↑; Compared with control, $p < 0.01$ | He et al. (2020) |
| | | | 4 weeks | | |
| Mouse | 19 months | Resistance exercise | 3 times/week, | Blood and muscle; ↑; Compared with the control, $p < 0.05$ | Kim et al. (2015) |
| | | | 12 weeks | | |
| Mouse | 5 weeks | Aerobic exercise | 5 times/week | Blood; ↑; Compared with the control and high-fat diet, $p < 0.05$ | Chou et al. (2023) |
| | | | 8 weeks | | |
| Mouse | 2.5–3 months | Swimming | 5 times/week | Hippocampal; ↑; Exercise + AβOs[3] Compared with the AβOs, $p < 0.05$ | Lourenco et al. (2019) |
| | | | 5 weeks | | |
| Mouse | 6 months | Swimming | 4 times/week | Blood; ↑; Compared with control, $p < 0.001$ | Zhou et al. (2022) |
| Rat | 12 months | | 12 weeks | | |
| Rat | – | Swimming | 5 days/week | Hippocampal; ↓; Ex compared with control group, $p < 0.001$ | Hegazy et al. (2022) |
| | | | 5 weeks | | |
| Rat | 8 weeks | Aerobic exercise | 2 times/week | Kidney: ↑ Compared with the WKY-S[4] group, $p < 0.05$ | Luo et al. (2023) |
| | | | 14 weeks | SHR-L[5], SHR-M[6], and SHR-H[7] kidneys Compared with SHR-S[8], $p < 0.05$ | |

| Table 1 (continued) | | | | | |
|---|---|---|---|---|---|
| Model | Age | Exercise intervention style | Exercise frequency | Irisin detection methods and changes | Author (year) |
| Rat | 20 months | Voluntary wheel running | 12 weeks | Blood, Cardiac and Liver; ↑; Compared with 24-month-old sedentary rats, $p < 0.05$ | *Belviranlı & Okudan (2018)* |

**Notes:**
[1] MCI, moderate continuous intensity.
[2] HIIT, high-intensity intermittent training.
[3] AβOs, Aβ oligomers.
[4] WKY-S, Wistar-Kyoto-sedentary group.
[5] SHR-L, spontaneously hypertensive rats with low-intensity aerobic exercise training.
[6] SHR-M, spontaneously hypertensive rats with medium-intensity aerobic exercise training.
[7] SHR-H, spontaneously hypertensive rats with high-intensity aerobic exercise training.
[8] SHR-S, spontaneously hypertensive rats-sedentary group.
Source: Lagzdina R, Rumaka M, Gersone G, Tretjakovs P. 2020. Circulating irisin in healthy adults: changes after acute exercise, correlation with body composition, and energy expenditure parameters in cross-sectional study. *Medicina (Kaunas)* 56 DOI 10.3390/medicina56060274.

the reported outcomes of acute exercise on irisin secretion (*Cho et al., 2021*). There was a study assessed irisin levels in 84 adults before and after acute aerobic exercise. Results indicated post-exercise irisin levels remained unchanged in 58%, decreased in 23%, and increased in 19% of participants, highlighting inter-individual variability in irisin response to exercise (*Lagzdina et al., 2020*). Table 1 presents a comparative analysis of irisin levels before and after exercise across the aforementioned studies. The specific irisin levels in circulation and muscle are important for understanding the potential impact of these changes. It is important to explore whether the observed increases in the irisin level are sufficient to promote beneficial effects (such as improvements in myocardial function), which have been associated with elevated irisin levels. Future studies should clarify these relationships and determine the clinical relevance of exercise-induced irisin secretion.

## Effect of long-term exercise on irisin secretion

Regular physical activity has many cardioprotective benefits, including anti-atherosclerotic, anti-arrhythmic, anti-thrombotic, and anti-ischemic effects (*Franklin et al., 2020*). Compared with acute exercise, the types of long-term exercise vary (*Amanat et al., 2020*; *Huh et al., 2014*; *Li et al., 2021*; *Nygaard et al., 2015*; *Pazokian, Amani-Shalamzari & Rajabi, 2022*). As mentioned before, obesity is negatively correlated with irisin levels. In a 12-week study, both MICT at 60% $VO_{2max}$ and HIIT at 100% $VO_{2max}$ effectively reduced weight and increased irisin levels in obese subjects (*D'Amuri et al., 2022*). Similarly, a 12-week RE intervention for older adults revealed that 40–80% of one-repetition maximum (1-RM) increased serum irisin levels, which were negatively correlated with reduced body fat (*Zhao et al., 2017*). However, in a 16-week RE program for obese older women, although the body composition improved, irisin levels remained unchanged (*Tibana et al., 2017*). *Pazokian, Amani-Shalamzari & Rajabi (2022)* found that circulating irisin levels were not significantly changed after 6 weeks of functional training or with blood flow restriction (50–80% arterial occlusion pressure) in elderly individuals. These results indicate that while long-term exercise may boost irisin levels, the effect is not consistent across all populations.

In terms of animal research, in spontaneously hypertensive rats that underwent 14 weeks of low-intensity (30–40% maximum exercise capacity (MEC)), moderate-intensity (45–55% MEC), and high-intensity (60–70% MEC) exercise, skeletal muscle as well as serum irisin pigment PGC-1α and FNDC5 levels increased significantly. Moreover, low- and medium-intensity exercise significantly ameliorated renal damage (*Luo et al., 2023*). Meanwhile, 1 month of moderate-intensity exercise training (75% $VO_{2max}$) in elderly mice with critical limb ischemia significantly increased PGC-1α/FNDC5/irisin expression and mitochondrial fission and mitophagy (*He et al., 2020*). In addition, *Chou et al. (2023)* showed that body weight was significantly decreased, serum FNDC5 level was increased, and the homeostasis model assessment of insulin resistance was improved after 8 weeks of aerobic exercise (70% $VO_{2max}$) in mice with high fat-diet-induced obesity. Swimming intervention for 5 weeks increased FNDC5/irisin levels in the hippocampus of mice with Alzheimer's disease. In Wistar rats, irisin level in the hippocampus increased, but no changes were observed in the serum and cerebrospinal fluid (*Hegazy et al., 2022*; *Lourenco et al., 2019*). The secretion of irisin induced by exercise is summarized in Table 2.

## CVD-protective mechanisms of exercise-regulated irisin

Exercise has been shown to improve cardiovascular function and is recommended for rehabilitation after cardiovascular events (*Fiuza-Luces et al., 2018*). A previous study demonstrated that 8 weeks of different intensity exercises (30–70% $VO_{2max}$) increased irisin levels and reduced the risk of cardiovascular death and all-cause death *via* the activation of the MAPK/AKT/STAT3 pathway (*Luo et al., 2023*). Exercise rehabilitation is the main treatment for HF. Studies have demonstrated that FNDC5 expression in the myocardium increased after exercise and that the expression of irisin in the myocardium was higher than skeletal muscle (*Aydin et al., 2014a*; *Kuloglu et al., 2014*). In addition, several studies have shown that exercise can ameliorate CVD by increasing irisin expression (*Liu, Wei & Wang, 2022*). Irisin secretion is influenced by various exercise-related factors, such as the intensity, type, duration, and frequency. Exercise has the potential to increase circulating irisin levels, thereby enhancing glucose tolerance, reducing insulin resistance, alleviating type two diabetes symptoms, improving endothelial function, and ultimately decreasing the risk of diabetes-related complications (*Liu, Wei & Wang, 2022*; *Lu et al., 2015*; *Zhu et al., 2015*). Many studies have assessed the mechanisms of exercise-induced irisin secretion to improve CVDs. In rats with MI, the serum level of irisin in the MI exercise group was approximately two-fold higher than that in the MI sedentary group, and it was believed that increased irisin level could delay myocardial necrosis or promote myocardial repair (*Hassaan et al., 2019*). *Li et al. (2021)* found that different exercise types and skeletal muscle electrical stimulation enhanced irisin/FNDC5 expression and activated the irisin/FNDC5-PINK1/Parkin-LC3/P62 pathway, which regulates mitophagy and autophagy. Resistance exercise, in particular, significantly suppresses oxidative stress in mice with myocardial infarction by activating this pathway, modulates mitochondrial autophagy, and improves cardiac function, while also considering the potential adverse effects of excessive autophagy on skeletal muscle

**Table 2 Effects of exercise-induced irisin secretion on cardiovascular disease.**

| Model | Age | Exercise intervention style | Exercise duration | Irisin detection methods and changes | Author (year) |
|---|---|---|---|---|---|
| Mouse | 8 weeks | Aerobic exercise | 60 mins | Myocardial; ↑; RE Compared with control, $p < 0.01$ | Li et al. (2021) |
| | | Resistance exercise | 8 rounds | | |
| Mouse | 8 weeks | Aerobic exercise | 60 mins | Kidneys ↑; ME[1] Compared with MI[2] groups, $p < 0.01$ | Wu et al. (2020) |
| Mouse | 8 weeks | Aerobic exercise | 60 mins | Liver ↑; ME Compared with MI groups, $p < 0.01$ | Wang et al. (2023) |
| Rat | 8 weeks | Aerobic exercise | 30 mins | Blood, Skeletal muscle and cardiac muscle; ↑; MI+Ex and MI+DHM[3] Compared with control, $p < 0.01$ | Hassaan et al. (2019) |
| Rat | 8 weeks | Aerobic exercise | 60 mins | Blood ↑; Compared with control, $p < 0.003$ | Seo et al. (2020) |
| | | | | Abdominal visceral fat and epididymal fat; ↑; Compared with control, $p < 0.05$ | |
| Mouse | 9 weeks | HIIT | 23 mins | Gastrocnemius; Blood; HIIT[4] and MICT[5] Compared with control, $p < 0.01$ and $p < 0.05$ | Wang et al. (2021) |
| | | MICT | 40 mins | | |
| Mouse | – | Aerobic exercise | 60 mins | Blood and Skeletal muscle; ↑; ME Compared with MI Sed, $p < 0.01$ | Ren et al. (2022) |
| Mouse | 5 weeks | Aerobic exercise | 60 mins | Blood and Heart; ↑; DOX+EXE[6] Compared DOX group, $p < 0.001$ | Pan et al. (2021) |
| | | | | DOX+EXE Compared vehicle group, $p < 0.001$ | |
| Human | 22.1 ± 2.8 years | Endurance exercise | 5 hours | Blood; ↑; Compared with pre-training, $p < 0.05$ | Huang et al. (2017), Ma et al. (2021) |

Notes:
[1] ME, myocardial infarction with aerobic exercise.
[2] MI, myocardial infarction.
[3] MI + DHM, myocardial infarction + dihydromyricentin.
[4] HIIT, high-intensity interval training.
[5] MICT, moderate-intensity continuous training.
[6] DOX + EXE, doxorubicin + exercise.

function (Li et al., 2021). In mice with MI, 6 weeks of MICT (65–70% $VO_{2max}$) activated FNDC5/irisin expression in the myocardium and specifically activated the PI3K/Akt signaling pathway within cardiac tissue, which is associated with muscle growth and hypertrophy in both skeletal and cardiac muscle. This, along with the promotion of M2 macrophage polarization, inhibited the inflammatory response in the liver after MI (Wang et al., 2023). In an acyltransferase 1 (ALCAT1) knockout mouse model, specifically chosen to elucidate the role of ALCAT1 in metabolic regulation and stress response, 6 weeks of moderate-intensity exercise (65–70% $VO_{2max}$) effectively upregulated irisin levels and suppressed ALCAT1 expression (Ren et al., 2022). This intervention was shown to mitigate oxidative stress and cellular apoptosis in the skeletal muscle of mice with MI, underscoring the significance of ALCAT1 in muscle health post-MI. Furthermore, irisin could attenuate doxorubicin-induced epithelial-to-mesenchymal transition by inhibiting reactive oxygen species-induced NF-κB-Snail activation (Pan et al., 2021). Moderate-intensity exercise (65–85% $VO_{2max}$) increased irisin levels via Akt and ERK1/2 signaling pathway activation to protect against cardio-cerebrovascular diseases (Li et al., 2017). Of note, exogenous irisin injections or exercise-induced irisin appears to have a protective effect on cardiovascular function. After 12 weeks of moderate-intensity exercise (65–70% $VO_{2max}$),

**Table 3 Possible mechanisms of the protective effect of exercise-induced irisin against cardiovascular disease.**

| Experiment mode | Exercise types | Possible mechanisms/signaling pathways | Protective effect | Author (year) |
|---|---|---|---|---|
| MI mouse model | Aerobic exercise / Resistance exercise | ↑ FNDC5/irisin-↑ PINK1/Parkin-LC3/P62 Pathway | Alleviates oxidative stress (inhibition of apoptosis) | Li et al. (2021) |
| MI mouse model | Aerobic exercise | ↑ FNDC5/irisin-AMPK-Sirt1-PGC-1α signaling pathway | | Wu et al. (2020) |
| MI mouse model | Aerobic exercise | ↑ FNDC5/irisin-↓ ALCAT1 | | Ren et al. (2022) |
| MI rat model | Swimming | ↑ FNDC5/irisin-↑ Nrf2 | | Bashar, Samir El-Sherbeiny & Boraie (2018) |
| Mouse | Aerobic exercise | ↑ FNDC5/irisin-↓ NF-κB-Snail | | Pan et al. (2021) |
| Mouse | Aerobic exercise | ↑ PPARγ/Pgc1α-Fndc5 | | Abedpoor et al. (2018) |
| SHR | Aerobic exercise | ↑ FNDC5/irisin-↓ MAPK and AKT- ↑ STAT3 | | Luo et al. (2023) |
| Mouse | Aerobic exercise | ↑ FNDC5/irisin-↑ Akt and ERK1/2 signaling pathways | Maintains mitochondrial function/structure, Suppresses mitochondrial apoptosis | Li et al. (2017) |
| Mouse | Aerobic exercise | ↑ FNDC5/irisin-↑ DRP1, PINK1 and LC3B | | He et al. (2021) |
| Human | Physical exercise | PGC-1α/FNDC5/Irisin pathway ACE2/Ang 1-7 axis | Anti-inflammatory effect Anti-fibrotic effect | De Sousa et al. (2021) |
| MI mouse model | Aerobic exercise | ↑ FNDC5/irisin-↓ PI3K/Akt/NF-κB signaling pathway | | Wang et al. (2023) |
| MI rat model | Aerobic exercise | ↑ FNDC5/irisin-↓ β-MHC ↑ FNDC5/irisin-↑ αSMA | Improves myocardial remodeling | Hassaan et al. (2019) |

abdominal visceral fat, epididymal fat, and total cholesterol levels were reduced while irisin levels increased and improved heart function in Sprague Dawley rats. Moreover, irisin levels were negatively correlated with abdominal visceral and epididymal fat and positively correlated with ejection fraction, fractional shortening, and cardiac output (*Seo et al., 2020*). In a study by the present authors, it was found that circulating irisin levels and the number and function of endothelial progenitor cells were significantly increased in individuals with obesity after 8 weeks of high-intensity training (90% of $HR_{max}$) and moderate-intensity exercise (60% of $HR_{max}$) with dietary restriction (*Huang et al., 2017*). Meanwhile, *Wang et al. (2021)* showed that the increased irisin levels in serum and gastrocnemius of $ApoE^{-/-}$ mice after 6 weeks of HIIT (4 sets of 5 × 10-s sprints with 20 s of rest) and MICT (40% of the determined maximal running speed) training could attenuate oxidative damage, thereby helping prevent atherosclerosis. Another study showed that 8 weeks of moderate-high intensity exercise (70% $VO_{2max}$) upregulated the PPARγ/PGC-1α-FNDC5 pathway in the gastrocnemius muscle and heart muscle of mice (*Abedpoor et al., 2018*). The role of exercise-induced irisin secretion in cardiovascular diseases and the potential mechanisms underlying the protective effects of

exercise-induced irisin on cardiovascular diseases are summarized in Tables 2 and 3, respectively.

## DISCUSSION

Both acute and long-term exercise influence irisin expression, a hormone linked to cardiovascular health. Acute exercise transiently increases circulating irisin levels, which typically return to baseline within 30 min of exercise (*Loffler et al., 2015*). This transient response may be a key mechanism through which acute exercise confers immediate cardiovascular benefits. Chronic exercise, however, has been suggested to enhance the metabolic dynamics of irisin, as it was shown that circulating irisin levels were selectively boosted in subjects (*Ma et al., 2021*). This long-term effect could potentially contribute to sustained cardiovascular benefits observed with regular exercise. A previous review suggested that acute exercise raises circulating irisin levels while chronic exercise enhances its metabolic dynamics and selectively boosts irisin levels (*Zunner et al., 2022*). Long-term RE training produced two different results: a significant increase or no difference with baseline (*Amanat et al., 2020*; *Tibana et al., 2017*; *Zhao et al., 2017*). *Amanat et al. (2020)* studied the effects of 12 weeks of aerobic exercise (60–75% $HR_{max}$), RE (75–80% 1 RM), and combined exercise (RE + aerobic exercise) on the serum level of irisin and found that except for RE, irisin level increased after all other exercise protocols. In another study, three RE protocols were designed: single RE (10 RM), 21 weeks of RE (10 RM), and endurance exercise + RE. The results showed that single endurance exercise or long-term endurance training alone or combined with RE did not increase either FNDC5 mRNA expression in skeletal muscle or irisin secretion in older men (*Pekkala et al., 2013*). These discrepancies may be attributed to the variability in intervention modalities, durations, and intensities, as well as other methodological differences. These factors highlight the need for standardized protocols to better understand the relationship among exercise, irisin secretion, and cardiovascular health. For example, many exercise types, such as voluntary wheel running, swimming training, running treadmill, climbing resistance ladder, and vibration exercise, were used in animal studies (*Belviranlı & Okudan, 2018*; *Kim et al., 2015*; *Li et al., 2021*; *Zhou et al., 2022*), and the intensity of intervention also varied. Even with moderate-intensity exercise, several different levels of intensity, ranging from 30% to 85% $VO_{2max}$, were employed. In addition, the intervention period was different for long-term exercise, ranging from 3 to 21 weeks. Moreover, the subjects in the studies were different, and in animal studies, different mouse models, such as those for MI, spontaneous hypertension, obesity, Alzheimer's disease, HF, and others, were used. Because the intervention protocol should fit the experimental requirements of different animal models, it is difficult to develop an universal standard for each experiment. With regard to humans, variations such as age, weight, and disease as well as the timing of blood collection after exercise (immediately, 30 min, 1 h, 24 h, or at other intervals) can lead to differing results. Therefore, it is important to stringently control varying conditions to decipher the mechanism of exercise-induced irisin secretion. Indeed, addressing these factors may help clarify the inconsistencies observed across studies. In Table 1, the effects of exercise intervention on irisin levels are summarized. Another possible reason to explain

these discrepancies is differences in the specificity of commercially available ELISA kits for measuring circulating irisin in humans (*Dinas et al., 2017*; *Ma et al., 2021*). Owing to species diversity, the effects of irisin observed in mice may not be observed in humans (*Ma et al., 2021*; *Ou-Yang et al., 2021*). For example, after long-term exercise training, irisin level increased in animals. However, this may not be replicated in humans. Different study populations may also induce different results and affect irisin levels. For example, physical fitness in older individuals differs from that in young adults, and people with obesity have differences in physical function compared with healthy individuals. In terms of CVDs, exercise can increase serum irisin levels and protect cardiovascular function *via* anti-inflammatory effects, alleviate oxidative stress, improve myocardial remodeling, and maintain mitochondrial function (*Hassaan et al., 2019*; *He et al., 2021*; *He et al., 2020*; *Luo et al., 2023*; *Qin et al., 2022*; *Wang et al., 2023*). The potential protective effects of exercise-induced irisin on CVDs are detailed in Table 2, outlining the proposed mechanisms by which irisin may exert its beneficial influence on the cardiovascular system. Likewise, Table 3 details the mechanisms through which irisin may exert its beneficial influence. Long-term exercise intervention is mainly used for CVD treatment. In this regard, *Wu et al. (2020)* found that 6 weeks of aerobic exercise inhibited oxidative stress-induced apoptosis in the impaired kidneys of mice with MI, partially by activating the FNDC5/Irisin-AMPK-Sirt1-PGC-1α signaling pathway and inhibiting the expression of lysocardiolipin acyltransferase 1. In another study, exercise activated the angiotensin-converting enzyme 2 (ACE2, the receptor for SARS-CoV-2 (*Gheblawi et al., 2020*)) pathway and cleaved angiotensin II (Ang II) to Ang1-7, leading to physiological changes that can regulate irisin expression and benefit the cardiovascular system. Furthermore, physical exercise may activate the PGC-1α/FNDC5/irisin pathway and the ACE2/Ang1-7 axis to prevent SARS-CoV-2 infection (*De Sousa et al., 2021*). Existing research on exercise-induced irisin regulation in CVDs has focused on MI, atherosclerosis of the coronary artery, myocarditis, arrhythmia, and hypertension. Some studies used irisin injection therapy to improve cardiovascular function, thereby significantly benefiting different CVDs (*Matsuo et al., 2015*; *Yan et al., 2022*). In hyperglycemic stress, irisin regulated mitochondrial function through the AMPK pathway, ultimately promoting cardiomyocyte survival (*Xin et al., 2020*). Moreover, irisin exhibited a therapeutic effect of MI in reducing myocardial cell apoptosis and fibrosis and promoting cardiac angiogenesis (*Liao et al., 2019*). Irisin also alleviated pressure overload-induced cardiac hypertrophy by inducing protective autophagy *via* the mTOR-independent activation of the AMPK-ULK1 pathway (*Li et al., 2018*). Exercise could significantly increase FNDC5 expression in the muscle, thereby increasing irisin secretion. Serum irisin may reflect the overall metabolic state of the body and FNDC5 expression in the muscle may more directly reflect the metabolic activity of the muscle tissue. Finally, many studies demonstrated that exercise increased skeletal muscle-derived irisin secretion, which is affected by the type, intensity, frequency, and duration of exercise, and suggested that irisin is a promising prospect for CVD prevention and treatment, however, further studies are still warranted.

## CONCLUSIONS AND PERSPECTIVES

This review summarized the multifaceted roles of exercise-induced irisin secretion in cardiovascular protection and its underlying mechanisms. Existing evidence suggests that exercise not only stimulates irisin secretion but also positively affects the cardiovascular system by activating various signaling pathways. In addition, the role of irisin in regulating body weight, preventing obesity, and improving glucose and lipid metabolism has been recognized. Furthermore, its potential as a biomarker for conditions such as congestive HF and MI offers new perspectives for the assessment and management of CVDs. Although the effects of exercise on irisin secretion have been explored, the precise mechanisms require further investigation. Thus, future research should focus on elucidating the mechanisms of exercise-induced irisin secretion to provide a strong theoretical foundation and practical guidance for leveraging irisin in the prevention and treatment of CVDs.

## ACKNOWLEDGEMENTS

We would like to thank MogoEdit for its English editing during the preparation of this manuscript.

### Funding

This study is supported by the Guangdong Basic and Applied Basic Research Foundation (No. 2023A1515012011), the Guangdong Scientific Research Platform and Projects for the Higher-educational Institution (2023ZDZX2033), the Scientific Research Project of Sports Bureau of Guangdong Province (GDSS2022N012), the Open Fund of the Guangdong Provincial Key Laboratory of Physical Activity and Health Promotion (2021B1212040014), and the Macao Science and Technology Development Fund (Project code: 002/2023/ALC). The funders had no role in study design, data collection and analysis, decision to publish, or preparation of the manuscript.

### Grant Disclosures

The following grant information was disclosed by the authors:
Guangdong Basic and Applied Basic Research Foundation: 2023A1515012011.
Guangdong Scientific Research Platform and Projects: 2023ZDZX2033.
Scientific Research Project: GDSS2022N012.
Open Fund of the Guangdong Provincial Key Laboratory of Physical Activity and Health Promotion: 2021B1212040014.
Macao Science and Technology Development Fund: 002/2023/ALC.

### Competing Interests

The authors declare that they have no competing interests.

## Author Contributions

- Wenhuang Guo conceived and designed the experiments, performed the experiments, analyzed the data, prepared figures and/or tables, authored or reviewed drafts of the article, and approved the final draft.
- Jianwei Peng conceived and designed the experiments, performed the experiments, authored or reviewed drafts of the article, and approved the final draft.
- Jiarui Su conceived and designed the experiments, performed the experiments, analyzed the data, authored or reviewed drafts of the article, and approved the final draft.
- Jingbo Xia analyzed the data, authored or reviewed drafts of the article, and approved the final draft.
- Weiji Deng performed the experiments, authored or reviewed drafts of the article, and approved the final draft.
- Peilun Li analyzed the data, prepared figures and/or tables, and approved the final draft.
- Yilin Chen analyzed the data, authored or reviewed drafts of the article, and approved the final draft.
- Guoqing Liu analyzed the data, prepared figures and/or tables, and approved the final draft.
- Shen Wang analyzed the data, prepared figures and/or tables, and approved the final draft.
- Junhao Huang performed the experiments, analyzed the data, prepared figures and/or tables, and approved the final draft.

## Data Availability

This is a literature review.

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
