# Peer review of "The role and underlying mechanisms of irisin in exercise-mediated cardiovascular protection"

_PeerJ, doi:10.7717/peerj.18413_

## Round 0.1 · original submission · Major Revisions

The databases used for the article search are described, but the keywords should be specified, and any excluded articles should be mentioned along with reasons for exclusions.
The manuscript should discuss why serum irisin concentrations are considered potential biomarkers or predictors of cardiovascular diseases, despite the lack of correlation with muscle concentrations, as highlighted in Shen et al. 2017.
The manuscript would benefit from a clearer distinction between serum and muscle concentrations of irisin or a discussion on the lack of studies addressing this issue.
The manuscript suffers from poor English language, leading to weak sentence structures, repetitive sections, and listing-style writing, making it hard for readers to follow the narrative. Please correct.
While a broad range of research is covered, some references are incorrectly formatted, and newer references could have been selected. The content in subsections is not concise and goes off on tangents.
Conclusions are weak and need more clear, succinct summaries.

Reviewer 1 ·

Basic reporting

The manuscript effectively presents the role of exercise-elevated irisin as a cardiovascular protector. The literature and references are appropriate throughout. The authors clearly highlight the mechanisms of irisin in exercise-mediated cardiovascular protection.

Experimental design

Although the databases used for the article search are described and personal biases were minimized, it is essential to specify the keywords used. Additionally, it should be mentioned whether all resulting articles were included or if some were excluded, along with the reasons for any exclusions.

Validity of the findings

The findings presented are intriguing. I recommend discussing why serum irisin concentrations are considered a potential biomarker or predictor of cardiovascular diseases, despite the lack of correlation with muscle concentrations. It is noted in lines 134 and 135 that "AHF patients with higher serum irisin levels had a higher mortality, providing evidence that serum irisin is a predictive biomarker for 1-year all-cause mortality in AHF patients (Shen et al. 2017)." This contradiction warrants further exploration.

Additional comments

The manuscript is interesting. It would benefit from a clearer distinction between serum and muscle concentrations of irisin. Alternatively, the discussion could highlight the lack of studies addressing this issue.

·

Basic reporting

There are issues with english language in this manuscript, leading to 'listing style' writing and repetiitve sections and phrases. This is one of the major pitfalls of this as it makes it hard for the reader to follow the story.
More concise language is rquired, throughout the document. Authours need to consolidate their points and bring the information together, effectively.

Please refer to attached document for more information.

Experimental design

There is a broad range of research looked at here.
Some references are not formatted correctly, and in some cases newer refernces could of been selected but weren't.
The review is organised well, however, the actual content in some of the subsections is not concise and clear enough, it makes the sotry long winded and at times goes off on tangents.

Please refer to attached document for more information.

Validity of the findings

There is a lack of scientific impact here, or, at least, its hard to interpret the impact here due to the writing style. Conclusions are weak and need to be vastly improved, with more clear, succinct summaries.

Additional comments

Please refer to attached document for more information.

---

## Round 0.2 · Minor Revisions

Please refer to the reviewer comments and cite tables and figures more often in the text.

Reviewer 1 ·

Basic reporting

The authors have made substantial improvements in the revised manuscript. Although the response letter does not clearly highlight the changes made, all points have been adequately addressed in the main manuscript file.
Therefore, I believe the manuscript is suitable for publication
Best regards!

Experimental design

no comment

Validity of the findings

no comment

·

Basic reporting

Vast improvements have been observed in basic reporting. Improved writing skills (English checked by someone, noted in rebuttal). See additional notes for more information

Experimental design

Clear study design now observed, with organised structure.
See additional notes for more information

Validity of the findings

Conclusions well stated. Literature well addressed and discussed.
See additional notes for more information

Additional comments

Second round of revisions, reviewer comments (This hass been attached as a PDF too):
Thank you for taking the time to address all the comments from the reviewer. Your hard work is appreciated and acknowledged!
This review, titled ‘The role and underlying mechanisms of irisin in exercise-mediated’, has been since revised by the authors. The manuscript now reads very well and is a great summary of the work.
Although prose is a little repetitive at times, it does flow well and tells a logical story. Subsections are helpful in jumping to the information you need, and are ordered effectively.
I do have a few comments to add, (please see major and minor comments), however these are easily rectifiable.
Major points:
As so much work has gone into making wonderful tables and figures, please reference them more in the main body of text. Will help direct the reader and provide more useful information.
Figure 1 says reference is taken from another source. Please check with editor / journal that no copyright rules are being broken (has permission been asked?).
Minor points:
29: ‘Cardiovascular diseases (CVDs) is a global burden of disease’, first few words don’t gel (grammar issue).
Try: ‘Cardiovascular diseases (CVDs) are a clinical burden globally’
30: maybe worthwhile mentioning average age of study participants i.e., ‘median age of 54.4’
35: intervention, not interventions
54: ‘skeletal muscle fiber type transformation’, do you mean ‘switching’ here?
71: check reference (typographical)
77-78: ‘important crucial’ delete one of these words (or add ‘and’ between the two)
80: may be interesting to include that ADAM10 cleaved FNDC5 in skeletal muscle (https://doi.org/10.1186/s13578-020-00413-3)
82-84: ‘Both FNDC5 and irisin were first discovered in the skeletal muscle and serum of humans, rabbits, and mice (Hofmann et al. 2014).’ potentially move to previous section of discovery?
125: already abbreviated myocardial infarction, so just use MI here
139: CVDs are highly correlated…
175: delete support (sentence doesn’t make sense)
176: ‘abnormal serum irisin levels can potentially be used’
210: remove full stop before Pekkala reference
210-212: Interesting point, I’m glad you included this. Are there any examples in humans whereby exercise was counter-intuitive to the beneficial release or effects of irisin? Or is this something only seen in mouse models?
From table 1 it seems apparent that in humans, blood irisin increases after all forms of exercise. Maybe worthwhile to make a note of this here
212: rephrase, i.e., Table 1 provides ….
220: grammar: effects
223: Be consistent with subscript for VO2max throughout manuscript
233: check formatting of ‘PGC-1α’
234: remove full stop before Luo reference
235: Be consistent with subscript for VO2max throughout manuscript
249: higher expression compared to what? Individual muscle weight? Potentially make clearer for reader
250: ‘several studies have shown that exercise can promote CVD treatment., maybe can be worded better, e.g., ‘several studies have shown that exercise can ameliorate CVD by …’
256: ‘In rats with…’
260: The autophagy component of this pathway (irisin/FNDC5-PINK1/Parkin-LC3/P62) can be deleterious to muscle if over activated, maybe worthwhile mentioning if authors of this paper saw it was activated to a beneficial level to improve regulatory mechanisms
264/273: Maybe worthwhile mentioning that AKT signalling is associated with muscle growth / hypertrophy in skeletal and cardiac muscle, may help to show why this pathway is important
290: double full stop noticed
293-294: sentence doesn’t make sense? Try ‘irisin, as it was shown that circulating levels were selectively boosted in subjects’
311: space required before ‘were employed’

---

## Round 0.3 · accepted · Accept

The authors addressed all comments.